# Self-Medication Behaviors of Chinese Residents and Consideration Related to Drug Prices and Medical Insurance Reimbursement When Self-Medicating: A Cross-Sectional Study

**DOI:** 10.3390/ijerph192113754

**Published:** 2022-10-22

**Authors:** Ziwei Zhang, Pu Ge, Mengyao Yan, Yuyao Niu, Diyue Liu, Ping Xiong, Qiyu Li, Jinzi Zhang, Wenli Yu, Xinying Sun, Zhizhong Liu, Yibo Wu

**Affiliations:** 1School of Public Health, Peking University, Beijing 100191, China; 2Institute of Chinese Medical Sciences, University of Macau, Macao 999078, China; 3School of Health Policy and Management, Chinese Academy of Medical Sciences & Peking Union Medical College, Beijing 100006, China; 4Faculty of Arts and Humanities, University of Macau, Macao 999078, China; 5International School of Public Health and One Health, Hainan Medical University, Haikou 571199, China; 6School of Humanities and Management, Jinzhou Medical University, Jinzhou 121001, China; 7School of Humanities and Social Sciences, Harbin Medical University, Harbin 150076, China; 8School of Foreign Languages, Weifang University of Science and Technology, Weifang 262700, China; 9School of Finance and Trade, Liaoning University, Shenyang 110036, China

**Keywords:** self-medication, OTC, drug prices, medical insurance reimbursement, economic factors

## Abstract

Background: Self-medication has become a common phenomenon. Economic factors are important factors that affect the self-medication of residents. This study aimed to investigate the current status of self-medication behaviors in China and explored the related factors affecting considerations associated with medical insurance reimbursement or drug price in self-medication. Methods: A national cross-sectional investigation was conducted among Chinese people over 18 years old under a multi-stage sampling method through a questionnaire, which includes demographic sociological characteristics, self-medication behaviors and scales. The Chi-square test was used to analyze whether the respondents consider medical insurance reimbursement or drug price as an important factor when purchasing over-the-counter (OTC) drugs. Logistic regression was used to examine the associated factors of considering medical insurance reimbursement or drug price. Results: In total, 9256 respondents were included in this study; 37.52% of the respondents regarded drug prices as an important consideration, and 28.53% of the respondents attached great importance to medical insurance reimbursement. Elderly respondents who lived in the central region, had medical insurance, and had lower levels of health literacy were more likely to consider the medical insurance reimbursement, while respondents with high monthly family income as well as students were less likely to consider the same issue (*p* < 0.05). Respondents settled in the central and western regions, students, those without fixed occupations, those who suffered from chronic diseases, or those with lower health literacy were more likely to consider drug prices, while the respondents with bachelor degrees, urban population and high per capita monthly income were less likely to consider the drug prices (*p* < 0.05). Conclusion: Self-medication behaviors with OTC drugs were prevalent in China, and consideration factors of medical insurance reimbursement or drug prices were related to socio-demographic characteristics and health literacy. There is a need to take measures to reduce the economic burden of self-medication, improve the health literacy of residents and strengthen public health education.

## 1. Background

The World Health Organization defines self-medication as the independent selection and use of medication by an individual to treat illnesses and symptoms of which he or she is aware, or the self-administration of medication prescribed by a physician to treat a chronic or re-occurring condition [1]. Self-medication is an important part of self-care, and as public income, education, and health literacy increase, the public gradually self-medicates as an important way to treat mild diseases [2]. The sudden outbreak of COVID-19 caused both an increasing difficulty of residents accessing medical care and a steep rise in healthcare expenditures, which has thus brought great challenges to the Chinese healthcare industry. To a certain extent, self-medication can alleviate the difficulty of medical care and can reduce its cost [3]. Various studies have shown that self-medication is a common behavior, with 32.5–81.5% of the world population having experienced self-medication [4]. Since the implementation of the Classification and Management of Prescription Drugs and Over-the-Counter Drugs (Trial) formulated by the State Drug Administration of China in 2000, China has entered the era of classification and regulation of prescription drugs (receptor, RX). In addition, over-the-counter (OTC) drugs have become increasingly important in promoting public health, and they gradually have gained greater potential social value in saving medical costs and relieving medical pressure [5]. However, while self-medication behaviors are developing, some accompanied inappropriate methods are also becoming emerging problems worldwide. Common inappropriate practices include short treatment duration and inadequate dosing [6]. Given the fact that China has approx. 2.5 million hospitalizations annually due to improper self-medication behaviors and 100,000 deaths from adverse drug reactions, the safety of self-medication behavior must be taken seriously [7].

The health expenditure of residents is related to many factors [8,9,10,11]. Demographic characteristics such as age, region, marriage and education level will have different impacts on medical expenditure [12,13,14]. Studies reported that education level, marital status, financial status and presence of health insurance are factors that influence the occurrence of self-medication behavior [15]. In recent studies, educational attainment, family income, and family structure independently influence factors on drug awareness [16]. Research shows that the impact of income on health expenditure is mostly positive [8]. Due to the different living costs of economic development in different regions, there are differences in health expenditure [9]. China’s research shows that the health expenditure in eastern and urban areas is higher, which may be because the medical and health resources and professionals in eastern and urban areas are more abundant than those in central areas and rural areas, providing more convenient and comprehensive health services; thus, the medical costs are higher [10]. The improvement of education level will improve people’s health awareness and make them more willing to pay more for health expenditure [17], which is consistent with the reason that people with high health literacy pay more attention to health promotion [18]. Self-medication belongs to the scope of public behavioral decision making, and personality traits can also influence public behavioral decision making [19]. One study showed that the main factor in taking advice from health professionals when purchasing OTC drugs was personality traits [20].

Economic factors have an important impact on self-medication behavior, including consideration of drug prices and medical insurance reimbursement. Basic medical insurance has huge impacts on health expenditure [11]. All countries in the world hope to reduce the medical burden of residents by improving medical insurance [21,22]. People with medical insurance are more likely to make full use of medical services [23]. In 1998, the Chinese government introduced basic medical insurance (UEBMI) for urban workers, followed by the new rural cooperative medical system (NCMS) for rural residents in 2003. In 2007, the urban residents’ basic medical insurance (URBMI) was introduced for children, students, the elderly, the disabled and other urban non-working populations [24]. In 2018, urban residents’ basic medical insurance and new rural cooperative medical insurance were merged into urban and rural residents’ medical insurance. In terms of underwriting services, the inpatient and outpatient services of urban workers’ insurance are the most comprehensive, while the urban and rural residents’ insurance mainly covers inpatient and part of chronic disease outpatient services [25]. The state has issued a series of policies to develop the medical insurance system, aiming to improve the accessibility and quality of drugs. In particular, the essential drug list (EDL) of drugs that are particularly important for national primary health care services was identified as a subset of the national reimbursement drug list [26]. In general, basic medical insurance for urban workers provides better outpatient and inpatient medical security than the basic medical insurance for urban and rural residents. The latter generally has limited coverage of outpatient treatment and a low reimbursement rate [27]. Ten years into the healthcare reform, China’s reform goals have been steadily promoted, and the residents’ equal access to medical care has been greatly improved, with those of lower socioeconomic status benefiting the most, but there is still much work to be carried out [28]. The price of drugs has always been one of the focuses of the medical and health industry. From 2014 to 2017, the proportion of drug expenses in total health expenditure in China were, respectively, 39.63%, 37.71%, 36.32% and 34.42%. Despite the overall downward trend, compared with the 10% to 20% proportion of drug expenses in developed countries, the proportion in China still ranks at a relatively high level [23]. Some studies have shown that the cost of self-medication is lower than going to the hospital, and the cost of self-medication is essentially the price of the drugs. Therefore, the price of drugs is an important factor influencing self-medication [29]. Therefore, reasonable regulation of drug prices and improvement of medical insurance are essential to ease the economic burden of individuals and even the country and to promote the health of the population.

Thus far, research on China’s self-medication behavior is still in the initial stages, and the types mainly focus on surveys of the current status of self-medication behaviors, investigation of self-medication knowledge, investigation of self-medication attitude, and analysis of factors influencing self-medication behaviors. There remains no detailed consideration of self-medication, which is where our study comes in. This study focuses on the economic factors that affect drug prices and medical insurance reimbursement during self-medication, including demographic characteristics, health literacy and personality characteristics. At the same time, according to research data, the relative increase in the number of Google searches for information on disease self-treatment worldwide since the COVID-19 pandemic implies an increase in the number of people searching for information on self-treatment for various diseases during the pandemic [30]. Especially in developing countries, a greater proportion of self-medication behavior occurs due to the prevalence of infectious diseases, especially antibiotics [31]. During the COVID-19 epidemic, residents’ self-medication purchase behavior and considerations changed. People are more sensitive and pay more attention to health-related information [32]. By using the latest data of China in 2021, this study analyzed residents’ self-medication behavior under the background of the COVID-19 epidemic and the influencing factors on drug prices and medical insurance reimbursement when purchasing drugs, aiming to help international readers better understand the influencing factors of Chinese residents’ self-medication and the reasons for paying attention to drug economic factors. It can better improve Chinese residents’ rational self-medication behavior to make conducive health decisions and provide reference for residents’ health management.

Thus, this study aims to investigate the current status of self-medication behavior of the Chinese public, explore the prevalence of self-medication and the types of OTC drugs purchased by the public, and identify the factors associated with the considerations of self-medication, including drug prices, drug reimbursement by medical insurance, advice from medical personnel, advice from family members, and advice from friends. Finally, this study further investigates the relevant influence factors with drug prices and medical insurance reimbursement as important considerations.

## 2. Materials and Methods

### 2.1. Study Design

The data for this study were obtained from the China Family Health Index—2021 (CFHI-2021). A multi-stage sampling method was used to gain samples in China mainland. Capital cities of all provinces/autonomous regions and direct municipalities were firstly selected as samples; then, other non-capital cities were selected using a random number table. Overall, there were 120 cities selected nationwide. The population of each city is stratified according to gender, age and urban–rural distribution, based on the data report of the “Seventh National Census in 2021” [33], which determines the number of samples at each level (per 100 people). The investigators in each city conducted convenience sampling subject to meeting quota requirements. The investigation was conducted from 10 July to 15 September 2021, and at least one investigator was recruited from each city.

The investigation was carried out through an online questionnaire platform called Wenjuanxing, the most popular investigation software in China (https://www.wjx.cn/ accessed on 18 October 2022), by investigators issuing questionnaires to residents one-on-one and face-to-face. Each investigator was responsible for collecting 30–90 questionnaires. Respondents answered by clicking on the link, and the questionnaire number was orderly provided by investigator. If the respondent could think but did not have enough action ability to answer the questionnaire, the investigator would conduct one-on-one questioning and answers instead. The study was approved by the Medical Ethics Committee of Jinan University (approval number: JNUKY-2021-018).

### 2.2. Participants

#### 2.2.1. Calculation of Minimum Sample Size

We used the following formula to calculate the minimum sample size [34],

n = [Z _α/2_^2^pq]/δ

In the above formula, n represents the sample size, p represents the estimated self-medication rate, q = 1 − p, α = 0.05, Z _α/2_ = 1.96 ≈ 2, δ is the permissible error, δ = 0.1 × p. According to literature reports, the self-medication rate of people around the world is about 32.5–81.5% [4]. The smaller value is used to calculate the sample size. The minimum sample size calculated by substituting the formula is 831. Considering an invalid questionnaire rate of 20%, the minimum number of distributed questionnaires should be 1039.

#### 2.2.2. Inclusion Criteria

(1) Age > 18;

(2) Have ever purchased and used OTC (if the answer was “yes” to the question of whether you have ever purchased and used OTC medications on your own);

(3) Participate in the study and fill in the informed consent form voluntarily;

(4) Participants can complete the network questionnaire investigation by themselves or with the help of investigators.

#### 2.2.3. Exclusion Criteria

(1) People with unconsciousness or mental disorders;

(2) People who are participating in other similar research projects;

(3) Medical staff. Since they have relatively specialized knowledge of medicine, they were excluded from this study, which aims to investigate the self-medication behavior of residents with no particular medicine backgrounds.

Following the above inclusion and exclusion criteria strictly, this study firstly excluded people 18 years old and below, and then medical workers. With 9344 people remaining, this study further excluded another 88 people who had not ever practiced self-medication. Finally, a total of 9256 cases were included. See Figure 1.

### 2.3. Instruments

The questionnaire consisted of four parts, focusing on the current status of residents’ self-medication behavior and related influencing factors. The first part investigated the social-demographic characteristics of the residents. The second part investigated the current status of residents’ self-medication behaviors and important considerations, including 3 questions (1 single-choice question, and 2 multiple-choice questions). The third part was a 10-item short version of the Big Five Inventory (BFI-10). The fourth part was the Short-Form Health Literacy Instrument (HLS-SF12). See Table 1 for details.

#### 2.3.1. The General Clinical and Demographic Information

The general demographic information of the respondents was asked through a self-designed questionnaire. The investigation included the gender, age, province, place of permanent residence (urban, rural), education level, per capita monthly income of the family, marital status, the current main way of bearing medical expenses, current occupational status (student, on-the-job, and no fixed occupation or retired), and currently diagnosed chronic diseases of the respondents.

#### 2.3.2. Items for Resident Self-Medication Status and Important Considerations

The first single-choice question was “Have you ever purchased and used OTC medicines on your own?”. Respondents who answered “No” to this question were excluded from the study.

The first multiple-choice question was “What kinds of OTC drugs have you ever purchased and used?”, and its 10 options were (1) antipyretic analgesics (e.g., paracetamol); (2) digestive system drugs (e.g., ranitidine hydrochloride capsules); (3) respiratory system drugs (e.g., aminocaffeine tablets); (4) vitamins/minerals (e.g., vitamin C tablets); (5) antibacterial drugs (e.g., metronidazole buccal tablets); (6) drugs for external use (e.g., compound beclomethasone camphor cream); (7) Chinese patent drugs (e.g., Xiao Chai Hu granules, Xiao Jianzhong granules, Sijunzi pills); (8) gynecological drugs (e.g., miconazole nitrate suppositories); (9) anti-allergic drugs (e.g., loratadine capsules); (10) others. For this multiple-choice question, the number of options available to the respondents could be 1–10. In the data analysis, for the respondents who selected the “other” option, the content filled in by the respondents was categorized, and a word frequency cloud map was drawn.

The second multiple-choice question was “Which of the following factors are important considerations when purchasing OTC drugs?”, and its 16 options were: (1) the price of the drug; (2) the efficacy of the drug; (3) the safety of the drug; (4) the taste of the drug; (5) brand awareness; (6) whether the drug can be reimbursed by medical insurance; (7) advice from medical staff (including doctors, pharmacists, etc.); (8) advice from family members; (9) advice from friends; (10) personal experience; (11) advertising; (12) after-sales service; (13) corporate reputation; (14) ease of taking the drug; (15) packaging of the drug; (16) dosage form of the drug. For this multiple-choice question, the number of options available to investigate respondents ranged from 1–16. The order in which the options appeared in the two multiple-choice questions was random for each respondent.

#### 2.3.3. The 10-Item Short Version of the Big Five Inventory (BFI-10)

A short version of the Big Five Inventory (BFI-10) was used to measure the personality traits of the respondents. The scale consists of 10 entries divided into five dimensions: extraversion, agreeableness, conscientiousness, neuroticism, and openness, with each dimension containing two entries. All questions on the scale were scored on a five-point Likert scale (1–5), and each subscale was scored out of 10, with higher scores indicating a more pronounced personality trait. Several studies have shown that the BFI-10 has good reliability and validity. In this study, referring to relevant literature, the five personality characteristics of the respondents were divided into a high group (7–10 points) and a low group (6 points and below).

#### 2.3.4. The Short-Form Health Literacy Instrument (HLS-SF12)

The health literacy of the respondents was measured by HLS-SF1230. The scale includes 3 dimensions of health care, disease prevention, and health promotion, with a total of 12 items, and each item is scored on a 4-point scale (1 = very difficult, 2 = difficult, 3 = easy, 4 = very easy). A Standardized HL index was calculated using a formula with an index range of 0–50, and its score was positively correlated with the health literacy of the respondents. The calculation formula is index = (mean − 1) × (50/3), where the mean is the average of all items involved in each individual, 1 is the minimum possible value of the mean (when the minimum value of the index is 0), 3 is the range of the mean, and 50 is the maximum value of the index. The higher the index, the higher the health literacy level of the investigation respondents. In the study, Cronbach’s coefficient of the scale was 0.940, and the Cronbach’s coefficients of the three subscales of health care, disease prevention and health promotion were 0.856, 0.860, and 0.868, respectively, with good reliability. In this study, the health literacy of the investigation respondents was divided into the high group (over 33 points) and low group (33 points and below) with reference to relevant literature [35,36].

### 2.4. Statistical Methods

Data entry and analysis were performed using SPSS™ for Windows (version 25.0) (SPSS Inc., Chicago, IL, USA). All scale scores are transformed into a binary variable (high-score and low-score grouping), with reference to relevant literature. Categorical variables were expressed by frequency (constituent ratios). The Chi-Square test was used for comparison between groups. Multivariate binary stepwise logistic regression was used to conduct a multi-factor analysis of the two variables of whether the investigation respondents considered health insurance reimbursement or drug price as an important consideration when purchasing OTC, with a test level of α = 0.05. The inclusion and exclusion criteria of variables were *p* = 0.05 and *p* = 0.10, respectively. Unless otherwise stated, the test level of statistical tests was α = 0.05. In the study, the independent variables included the demographic and sociological characteristics of the respondents and the score grading of the HLS-SF12, the BFI-10, and the EQ-5D-VAS scale. The two dependent variables were whether the respondents used Medicare reimbursement or drug price as an important consideration when purchasing OTC drugs on their own. Due to space constraints, we only discussed two economically relevant considerations when purchasing OTC drugs in this study, the correlation between the two dependent variables of health insurance reimbursement and drug prices. We will further investigate the correlates of the other dependent variables in subsequent studies.

### 2.5. Quality Control

The study conducted two pre-test rounds prior to the formal investigation. Trained investigators distributed questionnaires to respondents and registered their codes one-on-one and face-to-face. Every Sunday evening during the investigation process, members of the research group communicated with the investigators to summarize, evaluate and give feedback on the collected questionnaires. After the questionnaires were collected, two people conducted back-to-back logic checks and data screening. If singular values were found during data analysis, the original questionnaire needed to be found and checked with the investigator before proceeding to the next step of the analysis.

## 3. Results

### 3.1. Information of Self-Medication of Respondents

Common method bias shows that Harman’s single-factor method showed five factors with eigenvalues greater than 1, and the variance contribution rate of the first main factor was 34.98%, which did not exceed 40%, indicating that there was no common method bias [37].

The investigation showed that 99.06% (9256 of 9344) of Chinese adults have self-medication behaviors. The two most common types of OTC purchased were antipyretics (5421, 58.57%) and vitamins/minerals (4851, 52.41%). Some respondents also bought OTC drugs including external medicine for skin (3480, 37.60%), digestive system medication (3289, 35.63%), antibacterial drugs (3219, 34.78%), Chinese patent drugs (2435, 26.31%), and respiratory medicine (1649, 17.82%). In addition to the “Other” option, the last two categories of drugs purchased by themselves were gynecological drugs (1057, 11.42%) and antiallergic drugs (1322, 14.28%). Among the 167 people who chose “Other”, 30 people filled in the antihypertensive drugs (prescription drugs), 40 people filled in the cold and flu or antipyretic drugs, 21 people filled in the traditional Chinese medicine or Chinese patent medicines, 13 people filled in the topical drugs, and 16 people filled in the antibiotics (anti-inflammatory drugs) such as cephalosporins and sulphonamides (See Figure 2 for details).

### 3.2. The Score of Each Scale of the Respondents

The scores of respondents on the short version of the HLS-SF12, the BFI-10, are shown in Table 2. Since the scores on each scale do not satisfy the normal distribution, the median and upper and lower quartiles were used to describe the central tendency and dispersion of the scores of each scale. In total, 5942 respondents (64.20%) had high health literacy (score above 33 on the short version of the HLS-SF12), 3349 respondents (36.18%) with high extroversion, 5182 respondents (55.99%) with high agreeableness, 4757 respondents (51.39%) with high conscientiousness, 2257 respondents (24.38%) with high neuroticism, 3565 respondents (38.52%) with high openness. See Table 2.

### 3.3. Univariate Analysis of Respondents’ Considerations When Self-Purchasing OTC Drugs

Among the 9256 respondents who had self-medication behaviors, 4289 (46.34%) were male and 4967 (53.66%) were female, 4246 (45.87%) were 19–35 years old, 3935 (42.51%) were 36–59 years old, and 1075 (11.61%) were 60 years old and above; 6674 cases (72.10%) lived in urban, and 2582 cases (27.90%) lived in rural; the percentages of the respondents achieved high school/junior high school and below, college, bachelor’s degree, and postgraduate degree were 39.81%, 14.04%, 39.48%, and 6.67%, respectively.

Among 9256 respondents, the top three considerations when buying OTC drugs were medical personnel’s suggestions (7979, 86.20%), drug safety (5901, 63.69%) and drug efficacy (5492, 59.28%); 37.52% (3473, 37.52%) of the respondents regarded drug prices as an important consideration, and 28.53% (2641, 28.53%) of the respondents considered medical insurance reimbursement. A chi-square test was used to conduct a univariate analysis of whether Medicare reimbursement or drug price was an important consideration in the purchase of OTC drugs. The results showed that there were significant differences in considering Medicare reimbursement as an important factor when purchasing OTC drugs among respondents of different gender, ages, education levels, marital status, employment status, the main way of medical expenses borne, whether diagnosed with chronic disease, per capita monthly family income, conscientiousness grading, openness grading, and health literacy grading (*p* < 0.05). There are significant differences in taking drug price as an important consideration among respondents of different ages, education levels, location, place of residence, marital status, employment status, the main way of medical expenses borne, whether they had diagnosed chronic disease, per capita monthly income, and health literacy grading (*p* < 0.05). See Table 3.

### 3.4. Multivariate Binary Stepwise Logistic Regression Analysis of Two Factors of Medical Insurance Reimbursement or Drug Price

#### 3.4.1. Medical Insurance Reimbursement

A multivariate binary stepwise logistic regression analysis was carried out with whether the respondents considered medical insurance reimbursement as an important factor as the dependent variable. The demographic and sociological characteristics of the respondents and the grading of each scale scored as the independent variables. The Omnibus test result of the established model was *p* < 0.001, the −2 log-likelihood value was 10,830.071, and the Hosmer–Lameshaw test result was *p* = 0.872 > 0.05, indicating that the model was of good quality.

The multivariate binary stepwise logistic regression showed that age, location, employment status, the main way of medical expenses borne, per capita monthly family income, and health literacy grading were related to whether the respondents considered medical insurance reimbursement as an important consideration when purchasing OTC. (Table 4)

Compared to young adults aged 19–35, respondents aged 60 years and over were more likely to consider medical insurance reimbursement (OR = 1.605, 95% OR 1.314–1.960, *p* < 0.001). Compared to respondents in eastern China, respondents in the central region were more likely to consider medical insurance reimbursement (OR = 1.140, 95%OR 1.020–1.275, *p* = 0.021). Compared to the employed population, the students were less likely to consider reimbursement as an important factor (OR = 0.767, 95%OR 0.656–0.898, *p* = 0.001). Compared to paying for medical care out-of-pocket, respondents participating in residential health insurance (OR = 1.561, 95%OR 1.364–1.787, *p* < 0.001) and other (OR = 1.867, 95%OR 1.595–2.187, *p* < 0.001) were more likely to consider medical insurance reimbursement as an important factor. Respondents with high per capita monthly family income (OR = 0.849, 95% OR 0.736–0.981, *p* = 0.026) were less likely to consider medical insurance reimbursement compared to respondents with low per capita monthly income. Compared to high health literacy respondents, low health literacy respondents were more likely to consider medical insurance reimbursement (OR = 1.126, 95% CI 1.021–1.242, *p* = 0.018).

#### 3.4.2. Drug Price

Multivariate binary stepwise logistic regression analysis was carried out with whether the respondents took drug price as an important consideration factor as the dependent variable and the demographic and sociological characteristics of the respondents and the grading of each scale score as the independent variables. The Ominbus test result of the established model was *p* < 0.001, the −2 log-likelihood value was 12,019.557, and the Hosmer–Lameshaw test result was *p* = 0.694 > 0.05, indicating that the model was of good quality.

Multivariate binary stepwise logistic regression showed that education level, location, place of residence, employment status, whether they had diagnosed chronic disease, per capita monthly income, and health literacy grading were related to whether respondents considered drug price as an important consideration when purchasing OTC. Compared to high school, secondary school and below, those with a bachelor’s degree were less likely to consider drug price as an important factor (OR = 0.732, 95%CI 0.6494–0.826, *p* < 0.001). Compared to respondents in the east part of China, those in the central part of China (OR = 1.135, 95%CI 1.023–1.259, *p* = 0.017) and west part of China (OR = 1.251, 95%CI 1.124–1.393, *p* < 0.001) were more likely to consider drug price as an important factor. Compared to rural respondents, urban respondents were less likely to consider drug price as a factor (OR = 0.828, 95%CI 0.748–0.917, *p* < 0.001). Compared to employed respondents, students (OR = 1.327, 95%CI 1.177–1.495, *p* < 0.001) and those with no steady occupation (OR = 1.189, 95%CI 1.053–1.344, *p* = 0.005) were more likely to consider drug price as an important factor. Compared to respondents without chronic diseases, respondents with chronic diseases were more likely to consider the price of medicines (OR = 1.215, 95% CI 1.124–1.393, *p* < 0.001). Compared to respondents with low per capita monthly income, respondents with medium per capita monthly income (OR = 0.901, 95% CI 0.815–0.996, *p* = 0.042) and high per capita monthly income (OR = 0.838, 95% CI 0.730–0.961, *p* = 0.011) were less likely to consider the price of medicines as an important factor. Low health literacy respondents were more likely to consider drug price than high health literacy respondents (OR = 1.265, 95% CI 1.154–1.387, *p* < 0.001) (Table 5). 

## 4. Discussion

Self-medication is a common phenomenon that has made an important contribution to human health and well-being, and the ratio of self-medication with OTC is generally increasing [1]. Self-medication is common in China, with 99.06% of respondents self-medicating. Among the 167 people who chose “other” as the type of OTC drugs they purchased, 30 filled in the drugs were “antihypertensive drugs”, which actually belonged to prescription drugs. This reflects that residents may buy prescription drugs without a doctor’s prescription, and residents may not have a clear understanding of the concept and type of OTC drugs. An Italian study on the cognition of OTC medicines also showed that more than half of the respondents were confused about the meaning of “over-the-counter” drugs [38]. Self-medication should be guided correctly to play its role.

China’s medical insurance policy and drug price policy have an impact on residents’ self-medication behavior. In China’s current medical insurance reimbursement policy, OTC drugs are generally not included in the reimbursement scope. There are 2151 drugs included in China’s national medical insurance catalog in 2009, of which 556 products belong to the OTC drug market [39]. Some safe and effective OTC drugs that are not included in the medical insurance reimbursement directory have been used by residents for a long time, but the current policies fail to fully share the economic burden of residents buying OTC drugs. Drug price has always been one of the focuses in the medical and health field [40,41,42]. In 2015, the availability of drugs was promoted through the implementation of market-oriented drug prices [43]. China should speed up the reform of drug prices and supervision to ensure that residents use drugs.

### 4.1. Considerations

#### 4.1.1. Drug Price

In this study, 37.52% of the respondents took the drug price, an important economic factor in self-medication, as a crucial matter of consideration. Price may play a decisive role when purchasing occurs [44]. People with different demographic characteristics have different perceptions of the price of OTC drugs. A Finnish [45] study on the preference of college students for OTC drugs showed that students often use OTC drugs, especially painkillers, and pay more attention to the price. This study found that compared with the working population, students (OR = 1.327, *p* < 0.05) and people with unstable occupations (OR = 1.189, *p* < 0.05) are more likely to consider drug price as an important factor, due to affordability, which plays an important role in purchase decision. People in both occupations with low income, small budgets and independent economies are more sensitive to drug price budgets. Some studies [46] found that educational attainment had a positive impact on self-medication, with those with higher education having more comprehensive information on the rational use of OTC medicines. Compared to those in high school, technical secondary school and below, people with bachelor’s degrees were less likely to consider drug price as an important factor (OR = 0.732, *p* < 0.05). This is because people with higher education have more health-related knowledge; thus, they pay more attention to the quality and effect of drugs. Meanwhile, people with higher education obtain more financially advantageous careers in job hunting; thus, they are less restricted by price factors.

Similar studies have shown that [47] people with one or more chronic diseases have higher self-medication costs, and drug use and drug expenditure increase more rapidly with the prevalence of chronic disease. Respondents with chronic conditions were more likely to consider drug prices as a significant factor versus respondents without those conditions (OR = 1.215, *p* < 0.05). Therefore, patients with chronic diseases need to continuously bear high drug costs in the long run. They tend to control drug expenditures to achieve the goal of long-term health maintenance. The economic level of rural areas lags that of cities, and the per capita disposable income is low. Rural people are more concerned about drug price when self-medicating (*p* < 0.05).

#### 4.1.2. Medical Insurance Reimbursement

The availability of medical insurance is an important predictor of self-medication, and income and cost-sharing are closely related to OTC consumption [48]. Medical insurance reimbursement of drugs is an important consideration for most consumers to purchase OTC drugs. The number of respondents in this study who mainly regard medical insurance reimbursement as an important consideration during self-medication reached 2641, accounting for 28.53%.

As a part of self-care, self-treatment is the most important and frequently taken health maintenance measure for the elderly [49]. Compared with the young adults aged 19–35, the elderly over 60 years old are more likely to consider medical insurance reimbursement as an important factor (OR = 1.605, *p* < 0.05), because elderly patients usually have more medical needs than young people. Medical insurance reimbursement has a limited impact on the utilization of medical services of the elderly with heavy medical burdens, and the elderly with a high utilization rate of medical services may still be unable to afford medical expenses after reimbursement [50]. The centralized medical insurance plan in China has an unfair impact on the utilization of medical care for economically vulnerable elderly groups [51]). Previous studies [52,53,54] showed that medical care utilization, drug consumption and medical expenditure of the elderly were higher than those of the general population. The elderly are weaker, older, limited by their conditions, have a greater need for OTC drugs, use medicines more frequently and have a traditional consumer mindset, and they are more likely to consider purchasing OTC drugs through medical insurance. Compared with the working population, the student population is less likely to consider medical insurance reimbursement as an important factor (OR = 0.767, *p* < 0.05). The results of Iran’s study [55] showed that there was no statistical difference in self-medication behavior between uninsured students and insured students. For the student group, due to their low level of participation in medical insurance reimbursement and lack of relevant knowledge, they seldom consider medical insurance reimbursement when buying OTC drugs. In China, almost all of the population (more than 95% in 2013) is enrolled in the social medical insurance scheme [56]. The reimbursement rates of basic medical insurance for urban workers, basic medical insurance for urban residents and the new rural cooperative medical system are 72%, 50% and 40%, respectively [57]. Compared with those who mainly bore their medical expenses, the respondents who participate in residents’ medical insurance (OR = 1.561, *p* < 0.05) and others (OR = 1.867, *p* < 0.05) are more likely to consider medical insurance reimbursement as an important factor. Lack of health insurance may have a negative impact on medication compliance and disease treatment control [58].

#### 4.1.3. Comprehensive Drug Price and Medical Insurance Reimbursement

As important economic factors of self-medication, medical insurance reimbursement and drug prices affect the choice of OTC drugs at the same time. People with low monthly income and in different regions consider medical insurance reimbursement and drug prices as factors, which are attributed to the influence of the external economic environment and individual economic level. The central (OR = 1.135, *p* < 0.05) and western (OR = 1.251, *p* < 0.05) respondents took drug price as an important consideration, while the central respondents also considered medical insurance reimbursement (OR = 1.140, *p* < 0.05). Those with low per capita monthly income considered drug price and medical insurance reimbursement as important factors (*p* < 0.05). The per capita disposable income in the eastern region was higher than that in the central and western regions, and the economic situation may have an impact on the possibility that respondents consider drug prices and whether drugs are reimbursed by medical insurance as important considerations. This may be an important reason for the above situation.

Sufficient health literacy is considered to be a prerequisite for rational self-medication management. Health literacy is an intermediary factor in the causal relationship between education and health [59], especially affecting the health outcomes of patients with low education and low health literacy. Low health literacy is related to many adverse health outcomes, such as poor health management ability and major diseases. Compared with the respondents with high health literacy, low health literacy is more likely to consider medical insurance reimbursement and drug price as important factors (*p* < 0.05). People with high health literacy have stronger health beliefs and safety awareness. They pay attention to the indications of drugs and focus on drug safety and effectiveness. The low health literacy is due to a lack of health awareness and drug use behavior dominated by economic factors. Health education helps residents understand the risks and expected results of drug use and promotes the improvement of health literacy [60]. The importance given to economic factors such as health insurance reimbursement and drug prices in self-medication was influenced by personal economic conditions as reflected by demographic characteristics and the regional macroeconomic environment and was not significantly related to personality.

### 4.2. Suggestions

Financial conditions are important factors that affect patients’ self-medication. Studies have shown that patients may change medication choices and medication compliance due to economic burdens [61]. Expanding health insurance coverage and adjusting social health insurance policies to reduce patients’ inefficient and unsafe self-medication behaviors due to financial burden are key steps toward health equity. It is suggested not to take “whether or not it is OTC” as the key to determine whether a certain variety is included in the medical insurance reimbursement list. The varieties included in the catalogue of drugs for national basic medical insurance should be those that guarantee the safety and effectiveness of drug use and safeguard basic medical needs. Self-management behavior plays a vital role in the health of residents [62]. Changing the residents’ perception of self-medication, focusing on the safety and efficacy of OTC options, and promoting and valuing health education and the safe use of OTC drugs can contribute to the effective and efficient development of medical care.

Improving the health care system and controlling drug prices can enable patients to make the right choice to care for their health by obtaining the best access to effective treatment at a lower cost. Therefore, we propose the following suggestions to the health sector, drug manufacturers and dealers, medical personnel and the public. First, the medical insurance department further justifies and expands the reimbursement catalog of OTC drugs to effectively guarantee the medical care needs of all people seeking medical treatment. Second, the relevant regulatory authorities should strengthen the supervision of prescription drugs sales in pharmacies. Third, drug manufacturers and distributors should consider both profit plus pricing from a cost perspective and the value of patients’ needs and social benefits to determine the optimal pricing range. Fourth, medical personnel should improve their professional ethics from the perspective of safety and efficacy and avoid selling high-price but ineffective drugs due to extra kickbacks earned from drug sales. Finally, residents should improve their health literacy, make rational consumption when self-medicating, understand the indications and adverse reactions of OTC drugs, and use them with caution. In conclusion, determining reasonable drug prices and formulating a reasonable list of Medicare reimbursement drugs are conducive to improving drug accessibility, reducing social burden, and enabling patients to pay attention to the efficacy and safety of drugs on the basis of acceptable prices.

### 4.3. Advantages and Limitations of the Study

This study has several advantages. First, we used the data of a cross-sectional investigation across China in 2021 to obtain a wide range of representative data. In addition, the study combined the big five personality, health literacy and health-related quality of life theory to analyze the residents’ behavior in purchasing OTC drugs, which made an empirical contribution to the theory of economic factors affecting health behavior and expanded the new theoretical application value of economic factors and self-medication behavior. At the same time, targeted suggestions were given for the future practice of adjusting medical insurance policies and paying attention to drug use health education, and further practical contributions were made.

This study also has several limitations. First, the data were based entirely on self-reported questionnaires, which may be affected by social expectations, self-report errors and poor memory. Second, this study adopts a cross-sectional design, and the results were only used to explore the relevant factors of dependent variables, unable to reflect the change trend and law of events. Third, the participants of this study are all in China, which means that it can only represent Chinese residents, and therefore cannot be analyzed and generalized to other countries.

We used the latest available data (2021). Under the continuous influence of COVID-19, in the following years, the behavior characteristics of residents buying OTC and the important factors considered may change further, and data summary conclusions may need to be updated. At the same time, in the future, the influence mechanism of demographic characteristics, health literacy and personality characteristics of respondents on economic-related considerations when purchasing OTC drugs needs to be confirmed through longitudinal studies.

## 5. Conclusions

Self-medication is prevalent among adults in China. The two most common types of over-the-counter drugs that people buy and use are ① antipyretics and analgesics and ② vitamins/minerals. Whether drug expenses can be reimbursed by medical insurance as well as drug prices are important considerations when people buy OTC drugs.

The possibility that people take medical insurance reimbursement and drug prices as important considerations when purchasing OTC drugs is affected by their demographic sociological characteristics and health literacy. Respondents with poor personal economic conditions and regional macro-economy pay more attention to medical insurance reimbursement and drug prices, and respondents with low health literacy are also more likely to take medical insurance reimbursement and drug prices as important considerations. China’s relevant departments should further promote the adjustment of medical insurance policies, reduce the burden of residents’ self-medication costs, strengthen the supervision of drug prices, protect the accessibility of drugs for residents with low economic levels, and regulate residents’ self-medication behavior.

## Figures and Tables

**Figure 1 ijerph-19-13754-f001:**
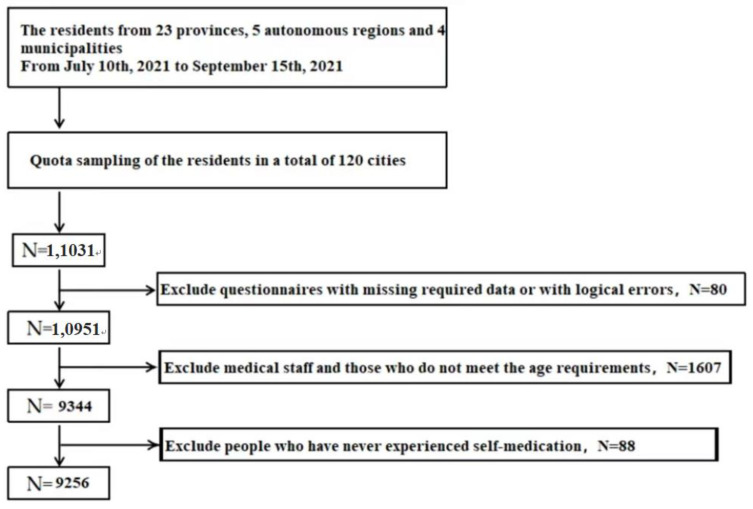
Flowchart of participant enrollment.

**Figure 2 ijerph-19-13754-f002:**
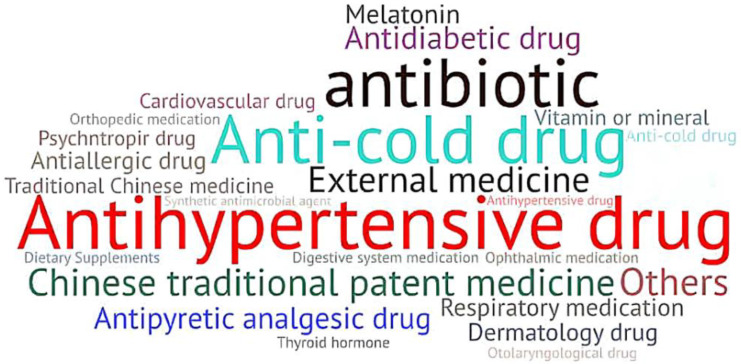
Word frequency map of “other” OTC medicines purchased and used by respondents.

**Table 1 ijerph-19-13754-t001:** Content and functions of the four parts of the questionnaire.

Part	Table Contents	Purpose
The general clinical and demographic information	Gender, age, province, place of permanent residence, education level, per capita monthly income of the family, marital status, the current main way of bearing medical expenses, current occupational status, and currently diagnosed chronic diseases	Collect general demographic information
Items for Resident Self-Medication Status and Important Considerations	This part includes 3 questions, “Have you ever purchased and used OTC medicines on your own?”, “What kinds of OTC drugs have you ever purchased and used?”, “Which of the following factors are important considerations when purchasing OTC drugs?”	Collect self-medication behavior and considerations
BFI-10	The scale consists of 10 entries divided into five dimensions: Extraversion, Agreeableness, Conscientiousness, Neuroticism, and Openness, with each dimension containing two entries	Measure the personality traits of the respondents
HLS-SF12	The scale includes 3 dimensions of health care, disease prevention, and health promotion, with a total of 12 items.	Measure the health literacy of the investigation respondents

**Table 2 ijerph-19-13754-t002:** Respondents’ scores on the HLS-SF12, BFI-10.

	No. of Items	Score Range	Kolmogorow–Smironov Z	*p* Value from the K-S Test	Median	Lower Quartile—Upper Quartile	High Score Group	Low Score Group
HLS—SF12	12	0–50	0.208	<0.001	33.33	30.56–37.50	5942 (64.20%)	3314 (35.80%)
BFI-10								
Extraversion	2	2–10	0.203	<0.001	6	5–7	3349 (36.18%)	5907 (63.82%)
Agreeableness	2	2–10	0.187	<0.001	7	6–8	5182 (55.99%)	4074 (44.01%)
Conscientiousness	2	2–10	0.200	<0.001	7	6–8	4757 (51.39%)	4499 (48.61%)
Neuroticism	2	2–10	0.217	<0.001	6	5–6	2257 (24.38%)	6999 (75.62%)
Openness	2	2–10	0.221	<0.001	6	6–7	3565 (38.52%)	5691 (61.48%)

**Table 3 ijerph-19-13754-t003:** Univariate analysis of whether the respondents consider drug Medicare reimbursement or drug price as an important consideration when purchasing OTC.

	Number of Respondents (%)	Medicare Reimbursement as an Important Consideration		*χ²* (*p*)	Drug Price as an Important Consideration		*χ²* (*p*)
		No	Yes		No	Yes	
Gender				**4.553 (0.033)**			1.321 (0.250)
Male	4289 (46.34)	3019 (70.4)	1270 (29.6)		2653 (61.9)	1636 (38.1)	
Female	4967 (53.66)	3596 (72.4)	1371 (27.6)		3130 (63)	1837 (37)	
Age(years)				**116.266 (0.001)**			**25.057 (<0.001)**
19–35	4246 (45.87)	3224 (75.9)	1022 (24.1)		2687 (63.3)	1559 (36.7)	
36–59	3935 (42.51)	2746 (69.8)	1189 (30.2)		2499 (63.5)	1436 (36.5)	
>60	1075 (11.61)	645 (60)	430 (40)		597 (55.5)	478 (44.5)	
Education level				**29.98 (<0.001)**			**82.066 (<0.001)**
High/Secondary School and lower	3685 (39.81)	2537 (68.8)	1148 (31.2)		2106 (57.2)	1579 (42.8)	
Junior college	1300 (14.04)	909 (69.9)	391 (30.1)		827 (63.6)	473 (36.4)	
Undergraduate	3654 (39.48)	2718 (74.4)	936 (25.6)		2460 (67.3)	1194 (32.7)	
Postgraduate degree (including master’s and PhD students)	617 (6.67)	451 (73.1)	166 (26.9)		390 (63.2)	227 (36.8)	
Location				4.952 (0.084)			**35.321 (<0.001)**
Eastern part of China	4722 (51.02)	3423 (72.5)	1299 (27.5)		3080 (65.2)	1642 (34.8)	
Central part of China	2391 (25.83)	1684 (70.4)	707 (29.6)		1459 (61)	932 (39)	
Western part of China	2143 (23.15)	1508 (70.4)	635 (29.6)		1244 (58)	899 (42)	
Place of residence				2.259 (0.133)			**62.524 (<0.001)**
Urban	1840 (19.88)	1816 (70.3)	766 (29.7)		1448 (56.1)	1134 (43.9)	
Rural	4472 (48.31)	4799 (71.9)	1875 (28.1)		4335 (65)	2339 (35)	
Marital status	2944 (31.81)			**105.565 (<0.001)**			**17.575 (0.001)**
Unmarried		3946 (68.4)	1819 (31.6)		3629 (62.9)	2136 (37.1)	
Married	6674 (72.1)	2399 (78.1)	673 (21.9)		1915 (62.3)	1157 (37.7)	
Divorce	2582 (27.9)	133 (68.9)	60 (31.1)		127 (65.8)	66 (34.2)	
Widowed		137 (60.6)	89 (39.4)		112 (49.6)	114 (50.4)	
Employment status	4735 (51.16)			**121.292 (<0.001)**			**77.424 (<0.001)**
Employed	3146 (33.99)	2883 (69.8)	1246 (30.2)		2751 (66.6)	1378 (33.4)	
Student	1375 (14.86)	1707 (79.6)	437 (20.4)		1327 (61.9)	817 (38.1)	
Unemployed		1533 (70.5)	641 (29.5)		1204 (55.4)	970 (44.6)	
Retired	5765 (62.28)	492 (60.8)	317 (39.2)		501 (61.9)	308 (38.1)	
The main way of medical expenses borne	3072 (33.19)			**105.436 (<0.001)**			**48.319 (<0.001)**
Out-of-pocket payments	193 (2.09)	1482 (80.5)	358 (19.5)		1100 (59.8)	740 (40.2)	
Resident Basic Medical Insurance (RBMI)	226 (2.44)	3163 (70.7)	1309 (29.3)		2693 (60.2)	1779 (39.8)	
Other		1970 (66.9)	974 (33.1)		1990 (67.6)	954 (32.4)	
Chronic diseases condition	4129 (44.61)			**50.086 (<0.001)**			**29.671 (<0.001)**
No	2144 (23.16)	5382 (73.2)	1975 (26.8)		4699 (63.9)	2658 (36.1)	
Yes	2174 (23.49)	1233 (64.9)	666 (35.1)		1084 (57.1)	815 (42.9)	
Monthly income (RMB)	809 (8.74)			**13.656 (0.001)**			**59.408 (<0.001)**
<4500		3388 (71.6)	1347 (28.4)		2783 (58.8)	1952 (41.2)	
4501–9000	7357 (79.48)	2194 (69.7)	952 (30.3)		2063 (65.6)	1083 (34.4)	
≥9001	1899 (20.52)	1033 (75.1)	342 (24.9)		937 (68.1)	438 (31.9)	
Extraversion				0.168 (0.682)			1.749 (0.186)
High-score Group	4735 (51.16)	2402 (71.7)	947 (28.3)		2122 (63.4)	1227 (36.6)	
Low-score Group	3146 (33.99)	4213 (71.3)	1694 (28.7)		3661 (62)	2246 (38)	
Agreeableness	1375 (14.86)			0.119 (0.731)			0.082 (0.774)
High-score Group		3696 (71.3)	1486 (28.7)		3231 (62.4)	1951 (37.6)	
Low-score Group	3349 (36.18)	2919 (71.6)	1155 (28.4)		2552 (62.6)	1522 (37.4)	
Conscientiousness	5907 (63.82)			**16.310 (<0.001)**			0.806 (0.369)
High-score Group		3312 (69.6)	1445 (30.4)		2993 (62.9)	1764 (37.1)	
Low-score Group	5182 (55.99)	3303 (73.4)	1196 (26.6)		2790 (62)	1709 (38)	
Neuroticism	4074 (44.01)			0.930 (0.335)			3.447 (0.063)
High-score Group		1631 (72.3)	626 (27.7)		1373 (60.8)	884 (39.2)	
Low-score Group	4757 (51.39)	4984 (71.2)	2015 (28.8)		4410 (63)	2589 (37)	
Openness	4499 (48.61)			**9.220 (0.002)**			3.059 (0.080)
High-score Group		2612 (73.3)	953 (26.7)		2267 (63.6)	1298 (36.4)	
Low-score Group	2257 (24.38)	4003 (70.3)	1688 (29.7)		3516 (61.8)	2175 (38.2)	
Health literacy	6999 (75.62)			**15.281 (<0.001)**			**69.020 (<0.001)**
High-score Group		4328 (72.8)	1614 (27.2)		3898 (65.6)	2044 (34.4)	
Low-score Group	3565 (38.52)	2287 (69)	1027 (31)		1885 (56.9)	1429 (43.1)	

bold font indicates statistical significance at 0.05 level.

**Table 4 ijerph-19-13754-t004:** Multi-factor binary stepwise logistic regression results with whether the respondents considered drug medical insurance reimbursement as an important factor as the dependent variable.

Variables	β	SE	Wald χ²	*p*	OR	The Lower Limit of 95%CI	The Upper Limit of 95%CI
Age (control group = 19–35)							
36–58	0.086	0.061	1.99	0.158	1.09	0.967	1.228
>60	0.473	0.102	21.45	<0.001	1.605	1.314	1.960
Location (control group = eastern part of China)							
Central part of China	0.131	0.057	5.359	0.021	1.140	1.020	1.275
Western part of China	0.115	0.059	3.769	0.052	1.121	0.999	1.259
Employment status (control group = employed)							
Student	−0.265	0.080	10.84	0.001	0.767	0.656	0.898
Unemployed	−0.001	0.071	0	0.985	0.999	0.87	1.147
Retire	0.087	0.104	0.703	0.402	1.091	0.89	1.339
The main way of medical expenses borne (control group = out-of-pocket payments)							
Resident Basic Medical Insurance (RBMI)	0.446	0.069	41.667	<0.001	1.561	1.364	1.787
Others	0.625	0.08	60.225	<0.001	1.867	1.595	2.187
Monthly income (RMB) (control group = 0–4500)							
4501–9000	0.066	0.053	1.549	0.213	1.068	0.963	1.186
≥9001	−0.163	0.073	4.951	0.026	0.849	0.736	0.981
Health literacy (control group = high health literacy)							
Low health literacy	0.119	0.05	5.628	0.018	1.126	1.021	1.242

β: Standardized regression coefficient; SE: Standard error; *p*: Statistical significance; OR: Odds ratio; 95% CI: 95% confidence interval.

**Table 5 ijerph-19-13754-t005:** Multi-factor binary stepwise logistic regression results with whether the respondents consider drug price as an important consideration as the dependent variable.

Variables	β	SE	Wald χ²	*p*	OR	The Lower Limit of 95%CI	The Upper Limit of 95%CI
Education level (Control group = high/secondary school and lower)							
Junior college	−0.106	0.071	2.239	0.135	0.9	0.783	1.033
Undergraduate	−0.311	0.061	25.688	<0.001	0.732	0.649	0.826
Postgraduate degree (including master’s and PhD students)	−0.007	0.099	0.005	0.944	0.993	0.818	1.206
Location (control group = eastern part of China)							
Central part of China	0.127	0.053	5.701	0.017	1.135	1.023	1.259
Western part of China	0.224	0.055	16.807	<0.001	1.251	1.124	1.393
Place of residence (control group = urban)							
Rural	−0.188	0.052	13.15	<0.001	0.828	0.748	0.917
Employment status (control group = employed)							
Student	0.283	0.061	21.565	<0.001	1.327	1.177	1.495
Unemployed	0.173	0.062	7.779	0.005	1.189	1.053	1.344
Retire	−0.041	0.086	0.228	0.633	0.96	0.811	1.136
Chronic diseases condition (control group = no)							
Yes	0.195	0.057	11.5	0.001	1.215	1.086	1.360
Monthly income (RMB) (control group = 0–4500)							
4501–9000	−0.104	0.051	4.143	0.042	0.901	0.815	0.996
≥9001	−0.177	0.07	6.425	0.011	0.838	0.730	0.961
Health literacy (control group = high health literacy)							
Low health literacy	0.235	0.047	25.074	<0.001	1.265	1.154	1.387

β: Standardized regression coefficient; SE: Standard error; *p*: statistical significance; OR: Odds ratio; 95% CI: 95% confidence interval.

## Data Availability

The original data of the paper can be obtained from the corresponding author upon reasonable request.

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
