# Peer review of "Self-Medication Behaviors of Chinese Residents and Consideration Related to Drug Prices and Medical Insurance Reimbursement When Self-Medicating: A Cross-Sectional Study"

_ijerph, 2022, doi:10.3390/ijerph192113754_

Round 1

Reviewer 1 Report

The title of this manuscript is interesting. However, the main problem at present is that the research purpose is not clear and the content presentation is rather confusing.

The correlation of survey scales used (such as the Health Literacy Scale, the Big Five Personality Scale, the EQ-5D visual simulation scale and the new general self-efficacy scale) and prices and medical insurance reimbursement of OTC drugs are not strong. It is recommended to reorganize the writing of the manuscript, and select the appropriate methods and contents related to the theme for writing.

Author Response

We were pleased to know that our manuscript entitled "What are the current status of self-medication behaviors of Chinese residents and the factors related to the consideration of drug prices and medical insurance reimbursement when self-medicating? A cross-sectional study (ijerph-1931728)" has received such valuable review comments.

I really appreciate all your insightful comments and suggestions, which have enabled us to grately improve our work. Based on the instructions provided in your letter, we uploaded the file of the revised manuscript. Accordingly, we have uploaded a copy of the original manuscript with all the changes highlighted in coloured text.

Appended to this letter is our point-by-point response to the comments and suggestions. The comments are reproduced and our responses are given directly afterward.

We are sure to have satisfactorily improved our manuscript and sincerely hope that it can be accepted for publication. Thanks again for the time and effort that you have put into reviewing our manuscript!

Yours Sincerely

Yi-bo Wu

October 3, 2022

Reviewer reports:

Reviewer 1

The title of this manuscript is interesting. However, the main problem at present is that the research purpose is not clear and the content presentation is rather confusing. The correlation of survey scales used (such as the Health Literacy Scale, the Big Five Personality Scale, the EQ-5D visual simulation scale and the new general self-efficacy scale) and prices and medical insurance reimbursement of OTC drugs are not strong. It is recommended to reorganize the writing of the manuscript, and select the appropriate methods and contents related to the theme for writing.

Response to comment:

Thank you for your suggestion.

  • To improve description of the purpose of the study, we have reorganized the writing of the manuscript, and the main purpose of our research is to analyze the factors affecting self-medication considering drug prices and medical insurance reimbursement. Based on the current situation and importance of self medication, the study explored residents' self medication considerations in the context of the current epidemic situation. Considering that economic factors are important considerations in drug purchasing, the study analyzed residents' considerations on drug prices and medical insurance reimbursement, and made suggestions for better targeted health education on self medication for the population.“By using the latest data of China in 2021, this study analyzed residents' self-medicationbehavior under the background of the COVID-19 epidemic and the influencing factors on drug prices and medical insurance reimbursement when purchasing drugs, aiming to help international readers better understand the influencing factors of Chinese residents' self-medication and the reasons for paying attention to drug economic factors, It can better improve Chinese residents' rational self-medication behavior to make conducive health decisions, and provide reference for residents’ health management.”
  • For the content presentation is more chaotic, the research ideas are rearranged. In the content presentation, the study first described the situation of self medication, including over-the-counter drugs commonly used by residents, then showed the health literacy of the subjects and the scores of the Big Five Personality Scale, and described the considerations of residents' self medication behavior. Finally, the main part of the study is to consider drug prices and medical insurance reimbursement for residents' self medication, and analyze the factors that affect residents' consideration of drug economic factors
  • The original survey scales used in the study included Health Literacy Scale, Big Five Personality Scale, EQ-5D Visual Simulation Scale and the New General Self efficacy Scale. Because the Big Five Personality Scale, EQ-5D Visual Simulation Scale and the new general self-efficacy scale are not closely related to OTC drug prices and medical insurance reimbursement, through literature review and considering that the theme of this study is to consider drug prices and medical insurance reimbursement when self medication, only the Health Literacy Scale and Big Five Personality Scale are retained for analysis. According to the literature review, previous studies have analyzed the influence of demographic characteristicssuch as age, region, marriage and education level, financial status and presence of health insurance [1-10]. Low health literacy adversely affects health outcomes in patients[11]. Research on Chinese population in 2022 shows that Big Five personality will affect residents' health care behavior[12], research also shows that Big Five personality is related to self-care behavior and health management[13]. Therefore, our study re-selected theme-related factors and reduced the scale into health literacy Scale and Big Five Personality scale. Previous literature review suggested that health literacy and personality traits were related to self-medication behavior. In the conclusion of this study, health literacy was related to the price of over-the-counter drugs and the consideration of medical insurance reimbursement. The economic factors considered by the respondents during self-medication were not strongly correlated with their personality traits. This suggests that economic considerations in self-medication are influenced by personal economic conditions and regional macroeconomic environment in demographic characteristics, as well as health literacy, but not the personality.And our study complements the review's view that personality traits play a role in listening to health care providers' recommendations when purchasing over-the-counter medications.

Furthermore, in order to show that the research is scientific and reasonable, the research also adds sample size calculation, common method deviation, and embellishes the article.

References

  1. Newhouse, J.P. Medical care costs: how much welfare loss? J Econ Perspect1992, 6, 3-21, doi: 10.1257/jep.6.3.3. 
  2. Li, L.; Du, T.; Hu, Y. The Effect of Population Aging on Healthcare Expenditure from a Healthcare Demand Perspective Among Different Age Groups: Evidence from Beijing City in the People's Republic of China. Risk Manag Healthc Policy2020, 13, 1403-1412, doi: 10.2147/RMHP.S271289. 
  3. Xu, X.; Wang, Q.; Li, C. The Impact of Dependency Burden on Urban Household Health Expenditure and Its Regional Heterogeneity in China: Based on Quantile Regression Method. Front Public Health2022, 10, 876088, doi: 10.3389/fpubh.2022.876088. 
  4. Sepehri, A.; Vu, P.H. Severe injuries and household catastrophic health expenditure in Vietnam: findings from the Household Living Standard Survey 2014. Public Health2019, 174, 145-153, doi: 10.1016/j.puhe.2019.06.006.
  5. Mackenbach, J.P.; Hu, Y.; Artnik, B.; Bopp, M.; Costa, G.; Kalediene, R.; Martikainen, P.; Menvielle, G.; Strand, B.H.; Wojtyniak, B.; et al. Trends In Inequalities In Mortality Amenable To Health Care In 17 European Countries. Health Aff (Millwood)2017, 36, 1110-1118, doi: 10.1377/hlthaff.2016.1674.  
  6. Mwabu, G.; Ainsworth, M.; Nyamete, A. Quality of medical care and choice of medical treatment in Kenya: an empirical analysis. J Human Res. 1993, 28, 838–62, doi: 10.2307/146295.
  7. Fuchs, V.R. Health care for the elderly: how much? Who will pay for it? Health Affairs(Millwood). 1999, 18, 11–21, doi: 10.1377/hlthaff.18.1.11.
  8. Aziz, M.M.; Masood, I.; Yousaf, M.; Saleem, H.; Ye, D.; Fang, Y.Pattern of medication selling and self-medication practices: A study from Punjab, Pakistan. PLoS One 2018, 13, e0194240, doi: 10.1371/journal.pone.0194240.
  9. Chen, S.; Yan, B. A study on the level of awareness and factors influencing parents' self-medication with antimicrobial drugs in children. Journal of the Army Medical University2022, 44, 960-966, doi:10.16016/j.2097-0927.202109200.
  10. Hjortsberg, C. Why do the sick not utilise health care? The case of Zambia. Health Econ.2003, 12, 755–70, doi: 10.1002/hec.839. 
  11. Qin, P.P.; Jin, J.Y.; Min, S.; Wang, W.J.; Shen, Y.W. Association Between Health Literacy and Enhanced Recovery After Surgery Protocol Adherence and Postoperative Outcomes Among Patients Undergoing Colorectal Cancer Surgery: A Prospective Cohort Study. Anesth Analg. 2022,134, 330-340, doi: 10.1213/ANE.0000000000005829.
  12. Zhang J, Ge P, Li X, Yin M, Wang Y, Ming W, Li J, Li P, Sun X, Wu Y. Personality Effects on Chinese Public Preference for the COVID-19 Vaccination: Discrete Choice Experiment and Latent Profile Analysis Study. Int J Environ Res Public Health. 2022, 15, 4842. doi: 10.3390/ijerph19084842.
  13. Skinner TC, Bruce DG, Davis TM, Davis WA. Personality traits, self-care behaviours and glycaemic control in type 2 diabetes: the Fremantle diabetes study phase II. Diabet Med2014, 31, 487-92. doi: 10.1111/dme.12339. 

Author Response

We were pleased to know that our manuscript entitled "What are the current status of self-medication behaviors of Chinese residents and the factors related to the consideration of drug prices and medical insurance reimbursement when self-medicating? A cross-sectional study (ijerph-1931728)" has received such valuable review comments.

I really appreciate all your insightful comments and suggestions, which have enabled us to grately improve our work. Based on the instructions provided in your letter, we uploaded the file of the revised manuscript. Accordingly, we have uploaded a copy of the original manuscript with all the changes highlighted in coloured text.

Appended to this letter is our point-by-point response to the comments and suggestions. The comments are reproduced and our responses are given directly afterward.

We are sure to have satisfactorily improved our manuscript and sincerely hope that it can be accepted for publication. Thanks again for the time and effort that you have put into reviewing our manuscript!

Yours Sincerely

Yi-bo Wu

October 3, 2022

Reviewer reports:

Reviewer 2

Generally speaking, this paper is well written, especially 2. Materials and Methods and 3. Results are very detailed. However, there are some problems in the writing method, especially the traces of Chinese writing style need to be corrected. In addition, the author's ability to summarize and summarize needs to be improved, which is most evident in the subtitle and discussion part. Of course, these are minor problems and do not involve the core defects. I believe that the author will gradually understand and improve in the future research and writing. Here are some specific ideas:

  • Title:Please write a new title. The number of words is controlled within 25, to be simple and clear, and can grasp the core of the full text, as far as possible to highlight the innovation of this article. Words like "the Current Status" should not be in the title.

Response to comment:

We feel great thanks for your professional review work on our article. We have revised the title of the article to"Self-Medication Behaviors of Chinese Residents and Consideration related to Drug Prices and Medical Insurance Reimbursement When Self-Medicating: A Cross-Sectional Study".

  • Abstract:The content should be simplified, especially the contents of Methods and Results are too long.

Response to comment:

Thanks. We have simplified the summary, focusing on the methods and results section

  • Background:
  1. Please add line numbers in the full text, so as to facilitate the reviewer to make modification suggestions.To inconvenience the reviewers is to inconvenience the authors themselves. But it's okay. I'm in a good mood today.

Response to comment:

Thank you for your advice, which has greatly benefited me. Line numbers and references have been added to the text.

  1. Thecontent of the literature review is too little, the reference is too few,only the secondparagraph.

Suggestions: First, add some new references appropriately, such as: The Impact of Dependency Burden on Urban Household Health Expenditure and Its Regional Heterogeneity in China: Based on Quantile Regression Method. Front. Public Health (2022) 10:876088. doi: 10.3389/fpubh.2022.876088. Second, some references inthe discussion section can be included.

Response to comment:

Thank you for your detailed advice. The background part is expanded from 616 words to 1292 words, Literature including "The Impact of Dependency Burden on Urban Household Health Expenditure and Its Regional Heterogeneity in China: Based on Quantile Regression Method. Front. Public Health (2022) 10:876088. doi: 10.3389/fpubh.2022.876088" been added. Firstly, the definition of self-medication behavior, current situation, adverse consequences and the importance of attention are introduced through combing. Then, according to the literature review, the influencing factors of self-medication behavior include demographic characteristics, health literacy and personality characteristics. Then, the influence of economic factors on self-medication behavior was introduced, including drug prices and medical insurance reimbursement. Finally, the current research status of self-medication was summarized. According to the suggestions on the discussion part mentioned later, the discussion of the study was integrated and modified. The references of the full text have been expanded from 44 to 64.

  1. In the last paragraph, a few sentences should be added to highlight the research innovation of this paper.

Response to comment:

Thank you for your advice. The current research has not yet analyzed the considerations of self medication in detail, but this research has refined the considerations of self medication, which is one of the innovations of our research. "This study focuses on the economic factors that affect drug prices and medical insurance reimbursement during the period of self medication, including demographic characteristics, health literacy and personality characteristics." At the same time, since the New Coronary Pneumonia pandemic, residents' self medication behavior and consideration factors have changed. This study, starting from the background of the New Coronary Pneumonia epidemic, understands residents' self medication behavior during the epidemic, which is another innovation of this study.

  • Materialsand MethodsThe content is very detailed, but it is suggested to add a table to summarize, so that readers can see more clearly, such as 2.3.3-2.3.6.

Response to comment:

Thank you for your advice.The Materials and Methods section has been refined. Table1 Contents and functions of the four parts of the questionnaire has been added to summarize. According to the literature review and re-sorting of research ideas, the scale used in this study has been changed to the health literacy Scale and the Big Five Personality Scale, and the scales with little correlation with self-medication behavior have been deleted. 2.1 Introduce the research design. 2.2 added the sample size calculation and retained the inclusion and exclusion criteria. 2.3 Survey tools introduced sociodemographic characteristics, self-medication related information collection methods, 2.3.3 design of the Big Five personality scale, 2.3.4 design of the health literacy scale. 2.4 is statistical method, 2.5 is quality control.

Table 1.  Contents and functions of the four parts of the questionnaire

Part

Table contents

Purpose

The general clinical and demographic information

Gender, age, province, place of permanent residence, education level, per capita monthly income of the family, marital status, the current main way of bearing medical expenses, current occupational status, and currently diagnosed chronic diseases

collect general demographic information

Items for Resident Self-Medication Status and Important Considerations

This part includes 3 questions, “Have you ever purchased and used OTC medicines on your own?”,“What kinds of OTC drugs have you ever purchased and used?”,“Which of the following factors are important considerations when purchasing OTC drugs?”.

Collect self medication behavior and considerations

BFI-10

The scale consists of 10 entries divided into five dimensions: Extraversion, Agreeableness, Conscientiousness, Neuroticism, and Openness, with each dimension containing two entries

measure the personality traits of the respondents

HLS-SF12

The scale includes 3 dimensions of health care, disease prevention, and health promotion, with a total of 12 items.

measure the health literacy of the investigation respondents

S1 The contents of Materials and Methods

Original

New

2.1

Study design

Polish the language

2.2

Inclusion criteria

Add:

Calculation of minimum sample size

2.3

Instruments

Delete:

2.3.5. The European Five-dimensional Health Scale (EQ-5D-5L) and 2.3.6. The New General Self-Efficacy Scale (NGSES)

2.4

Statistical methods

Polish the language

2.5

Quality control

Polish the language

  • Results
  1. 1 and 3.2 May be considered combined.

Response to comment:

Thank you for your advice. 3.1 and 3.2 has been combined. 3.1 The main display results are the original 3.2, showing the current situation of self medication and common OTC drugs. According to the following modification opinions, the original Table 1 in Part 3.1 is too simple, so the original Table 1 is integrated with the original Table 3 and Table 4 of single factor analysis, and the new Table 3 is integrated for display

  1. In addition, Table 1 is too simple. Considering that Table 3 and Table 4 contain information such as the number of visitors, I suggest that Table 1 be deleted or one or two graphs can be used here for simple explanation, and the variable names in Table 1 should be included in Chapter 2.

Response to comment:

Thank you for your advice. The results have been reorganized to retain the word frequency chart, health literacy and Big Five personality scale scores, univariate analysis and multivariate analysis. The original table 1 was removed and the common method bias added.Results Section 3.1 supplemented the description of common method deviation, 3.2 introduced the self medication of the respondents, 3.3 described the scores of various scales of the respondents, 3.4 the self purchase and use of over-the-counter drugs by the respondents, and the single factor analysis considering drug prices and medical insurance reimbursement was to delete the demographic characteristics of the original Table 1. Demographic information and single factor analysis results were displayed after merging the single factor analysis of the original Table 3 and Table 4.

S3  The contents of Results

Original

New

Difference

3.1

Demographic and sociological characteristics of the respondents

(Table 1. Demographic and sociological characteristics of respondents.)

Information of self-medication of Respondents

(Figure 2. Word frequency map of “other” OTC medicines purchased and used by respondents.)

Add:common method biases

New 3.1 mainly displays the original 3.2

Delete:

Original Table 1.

The contents of the original Table 1 are displayed in the new Table 3

3.2

Important Considerations for Respondents When Self-Purchasing OTC Drugs

(Figure 2. Word frequency map of “other” OTC medicines purchased and used by respondents.)

The score of each scale of the respondents

(Table 2. Respondents’ scores on the HLS-SF12, BFI-10.)

The original 3.2 is divided into the new 3.1 Self medication status and the new 3.3 Self medication considerations.

The original Table 2 is deleted as the new Table 2, which shows the scores of the Health Literacy and Big Five Personality Scale

3.3

The score of each scale of the respondents

(Table 2. Respondents’ scores on the HLS-SF12, BFI-10, EQ-5D-VAS scale and GSES.)

Univariate analysis of respondents’ considerations When Self-Purchasing OTC Drugs

(Table 3. Univariate analysis of whether the respondents consider drug Medicare reimbursement or Drug price as an important consideration when purchasing OTC.)

Integrate the original 3.4 single factor analysis and 3.1 basic information into the new 3.3

3.4

Univariate analysis of respondents’ considerations for purchasing OTC drugs

(Table 3. Results of the chi-square test on whether the respondents consider drug Medicare reimbursement as an important consideration when purchasing OTC.

Table 4. Results of the chi-square test on whether the price of the drug is an important consideration for respondents when purchasing OTC drugs.)

Multivariate binary stepwise logistic regression analysis of two factors of medical insurance reimbursement or drug price

(Table 4. Multi-factor binary stepwise logistic regression results with whether the respondents considered drug medical insurance reimbursement as an important factor as the dependent variable.

Table 5. Multi-factor binary stepwise logistic regression results with whether the respondents consider drug price as an important consideration as the dependent variable)

The original 3.5 is the new 3.4

3.5

Multivariate binary stepwise logistic regression analysis of two factors of medical insurance reimbursement or drug price(Table 5, Table 6)

  • DiscussionThe content of this part is very rich, but the writing method belongs to the typical Chinese style, which needs to make some adjustments. Advice:
  1. Section 4.1 can be simplified to a single paragraph as an opening paragraph, without the need to include too much outcome data.

Response to comment:

Thank you for your advice. 4.1 has been simplified to briefly introduce the current status of self-medication behavior in the study.

  1. The subheadings must be revised. It can't all be "Current Situation"... ", but should directly reflect the meaning of the topic.

Response to comment:

Thank you for your advice. The subtitle is summarized and modified according to the main content of this part.

  1. Section 4.3 is too long and has too many paragraphs. It is suggested to break it into several sections and show the core meaning of each section in the title. Suggestions: Use Firstly, secondly... Express the content more clearly.

Response to comment:

Thank you for your advice. 4.3 The discussion has been divided into several paragraphs, and the "First, second..." demarcation words have been used.

  1. Advantages and Limitations of the StudyIt's worth praising about limitations, but don't write too many about limitations. Try to limit it to about 2 items. Don't you just get yourself into trouble by writing too much? The tendency to be seized uponby others negates the originality of your research.

Response to comment:

Thank you for your detailed advice. The limitations of the research have been integrated. The limitations of the research methods, the research design, the research itself, and the limitations of the scope of applicability of the conclusions drawn from the Chinese subjects have been discussed

Furthermore, in order to show that the research is scientific and reasonable, the research also adds sample size calculation, common method deviation, and embellishes the article.

Reviewer 3 Report

The work is interesting and and very well written.The work is easy to read. The methodology is adequate. This study aimed to investigate the  current status of self-medication behaviors in China and explore the related considerations and explore the factors associated with medical insurance reimbursement or drug price in self-medication.  Thus, the topic is very important and relevant to Chinese society. However, after reading this article carefully, I have some opinions and suggestions for the author's reference:

1. I recommend expanding the conclusion which in its present form is very "poor".

2. Please provide additional research limitations and suggestions for future research.

Author Response

We were pleased to know that our manuscript entitled "What are the current status of self-medication behaviors of Chinese residents and the factors related to the consideration of drug prices and medical insurance reimbursement when self-medicating? A cross-sectional study (ijerph-1931728)" has received such valuable review comments.

I really appreciate all your insightful comments and suggestions, which have enabled us to grately improve our work. Based on the instructions provided in your letter, we uploaded the file of the revised manuscript. Accordingly, we have uploaded a copy of the original manuscript with all the changes highlighted in coloured text.

Appended to this letter is our point-by-point response to the comments and suggestions. The comments are reproduced and our responses are given directly afterward.

We are sure to have satisfactorily improved our manuscript and sincerely hope that it can be accepted for publication. Thanks again for the time and effort that you have put into reviewing our manuscript!

Yours Sincerely

Yi-bo Wu

October 3, 2022

Reviewer reports:

Reviewer 3

  • I recommend expanding the conclusion which in its present form is very "poor".

Response to comment:

Thanks. The conclusion part is expanded. Self-medication is prevalent among adults in China. “The two most common types of over-the-counter drugs that people buy and use are antipyretics and analgesics and vitamins/minerals. Whether drug expenses can be reimbursed by medical insurance and drug prices are important considerations when people buy OTC drugs. The possibility that people take medical insurance reimbursement and drug prices as important considerations when purchasing OTC drugs is affected by their demographic sociological characteristics and health literacy. Respondents with poor personal economic conditions and regional macro-economy pay more attention to medical insurance reimbursement and drug prices, and respondents with low health literacy are also more likely to take medical insurance reimbursement and drug prices as important considerations. “Put forward suggestions according to the research results. “China's relevant departments should further promote the adjustment of medical insurance policies, reduce the burden of residents' self-medication costs, strengthen the supervision of drug prices, protect the accessibility of drugs for residents with low economic levels, and regulate residents' self-medication behavior."

  • Please provide additional research limitations and suggestions for future research.

Response to comment:

Thank you for your advice. Limitations of the study and recommendations for future research have been revised. The limitations of the research are discussed separately from the limitations of the research method as a questionnaire, the limitations of the research design as a cross-sectional study, and the limitations of the conclusion drawn by the study population as Chinese population. For future research scenarios, first of all, under the background of COVID-19, the behavioral characteristics and important factors considered by residents in purchasing OTC may further change in the next few years, and the conclusion of data aggregation needs to be updated. Longitudinal studies are also needed to explore the arguments.

Furthermore, in order to show that the research is scientific and reasonable, the research also adds sample size calculation, common method deviation, and embellishes the article.

Round 2

Reviewer 2 Report

The author has addressed most of my questions. There is still one small problem to be solved:

Please merge 4.1 and 4.2 and delete two subheadings: 4.1. Current Situation of Self-Medication Behavior of Chinese Residents and 4.2. Medical insurance reimbursement policy and drug expenses in China.

In addition, please re-write the subheadings of 4.3.1-4.3.3, because "Consideration of..." It's too Chinglish. It's not academic enough.

Author Response

We were pleased to know that our manuscript entitled "What are the current status of self-medication behaviors of Chinese residents and the factors related to the consideration of drug prices and medical insurance reimbursement when self-medicating? A cross-sectional study (ijerph-1931728)" has received such valuable review comments.

I really appreciate all your insightful comments and suggestions. I'm glad that the previous revisions have completed most of the problems. These comments and suggestions enable us to improve our work gratefully. Enclosed are our itemized responses to comments and suggestions.

We are sure to have satisfactorily improved our manuscript and sincerely hope that it can be accepted for publication. Thanks again for the time and effort that you have put into reviewing our manuscript!

Yours Sincerely

Yi-bo Wu

October 16, 2022

Reviewer reports:

The author has addressed most of my questions. There is still one small problem to be solved:

  • Please merge 4.1 and 4.2 and delete two subheadings: 4.1. Current Situation of Self-Medication Behavior of Chinese Residents and 4.2. Medical insurance reimbursement policy and drug expenses in China.

Response to comment:

Thank you for your suggestion. We have merged 4.1 and 4.2 into the first part of the discussion, introduced the current situation of self medication research and current relevant policies in China, sorted out the logic and content. Two subheadings were deleted at the same time.

(2)In addition, please re-write the subheadings of 4.3.1-4.3.3, because "Consideration of..." It's too Chinglish. It's not academic enough.

Response to comment:

Thank you for your detailed advice. We have simplified and rewritten the subheadings of 4.3.1-4.3.3. After modification, it is 4.1. Considerations, 4.1.1 Drug price, 4.1.2 Medical insurance, reimbursement, 4.1.3 Comprehensive drug price and medical insurance reimbursement
